# Histone 4 lysine 5/12 acetylation enables developmental plasticity of *Pristionchus* mouth form

Michael S. Werner [1,2], Tobias Loschko[1], Thomas King[2], Shelley Reich [2], Tobias Theska [1], Mirita Franz-Wachtel[3], Boris Macek [3] & Ralf J. Sommer [1] ✉

Development can be altered to match phenotypes with the environment, and the genetic mechanisms that direct such alternative phenotypes are beginning to be elucidated. Yet, the rules that govern environmental sensitivity vs. invariant development, and potential epigenetic memory, remain unknown. Here, we show that plasticity of nematode mouth forms is determined by histone 4 lysine 5 and 12 acetylation (H4K5/12ac). Acetylation in early larval stages provides a permissive chromatin state, which is susceptible to induction during the critical window of environmental sensitivity. As development proceeds deacetylation shuts off switch gene expression to end the critical period. Inhibiting deacetylase enzymes leads to fixation of prior developmental trajectories, demonstrating that histone modifications in juveniles can carry environmental information to adults. Finally, we provide evidence that this regulation was derived from an ancient mechanism of licensing developmental speed. Altogether, our results show that H4K5/12ac enables epigenetic regulation of developmental plasticity that can be stored and erased by acetylation and deacetylation, respectively.

Different environments can elicit distinct phenotypes from a single genotype, referred to as phenotypic plasticity[1,2]. Ecological and theoretical approaches over the last 50 years have formalized the evolutionary implications and significance of plasticity[3–6]. More recently, molecular approaches are homing in on the mechanisms that direct environmental influence[7,8]. In contrast, the mechanisms that provide environmental sensitivity remain unknown. To do so requires an experimentally-tractable system of plasticity that is capable of linking environmental sensitivity, development and gene regulation.

*Pristionchus pacificus* has a short life cycle and reproduces primarily as self-fertilizing hermaphrodites similar to *Caenorhabditis elegans* (Fig. 1a), allowing the use of forward and reverse genetic tools[9]. Unlike *C. elegans* however, *P. pacificus* exhibits mouth-form plasticity. Specifically, adult worms express either a narrow Stenostomatous (St) morph with a single dorsal 'tooth', or a wide Eurystomatous (Eu)

morph with two teeth[10] (Fig. 1b). Morphological plasticity is coupled to behavioral plasticity, as St animals are strict bacterial feeders, while Eu animals can use their opposing teeth to kill other nematodes for food or competitive advantage[11–16]. This binary readout in laboratory model organism has enabled the discovery of a gene regulatory network for each morph[17–20]. In contrast, if and how epigenetic processes are involved in the regulation of plasticity remain poorly understood.

Here, we provide evidence that acetylation of histone 4 lysines 5/ 12 (H4K5/12ac) enables plasticity to different culture environments. Moreover, pharmacologically preventing deacetylation enforced epigenetic memory of a previous environmental experience; effectively canalizing a developmental trajectory despite changing environments. While several post-translational modifications to histone 3 have been implicated in dynamic gene regulation[21,22], outside of H4K16ac, the function of specific modifications on H4 has been less clear. The role of

[1]Department for Integrative Evolutionary Biology, Max Planck Institute for Biology Tübingen, Tübingen 72076, Germany. [2]School of Biological Sciences, The University of Utah, Salt Lake City, UT, USA. [3]Proteome Center Tübingen, University of Tübingen, Tübingen 72076, Germany. ✉e-mail: Ralf.Sommer@tuebingen.mpg.de

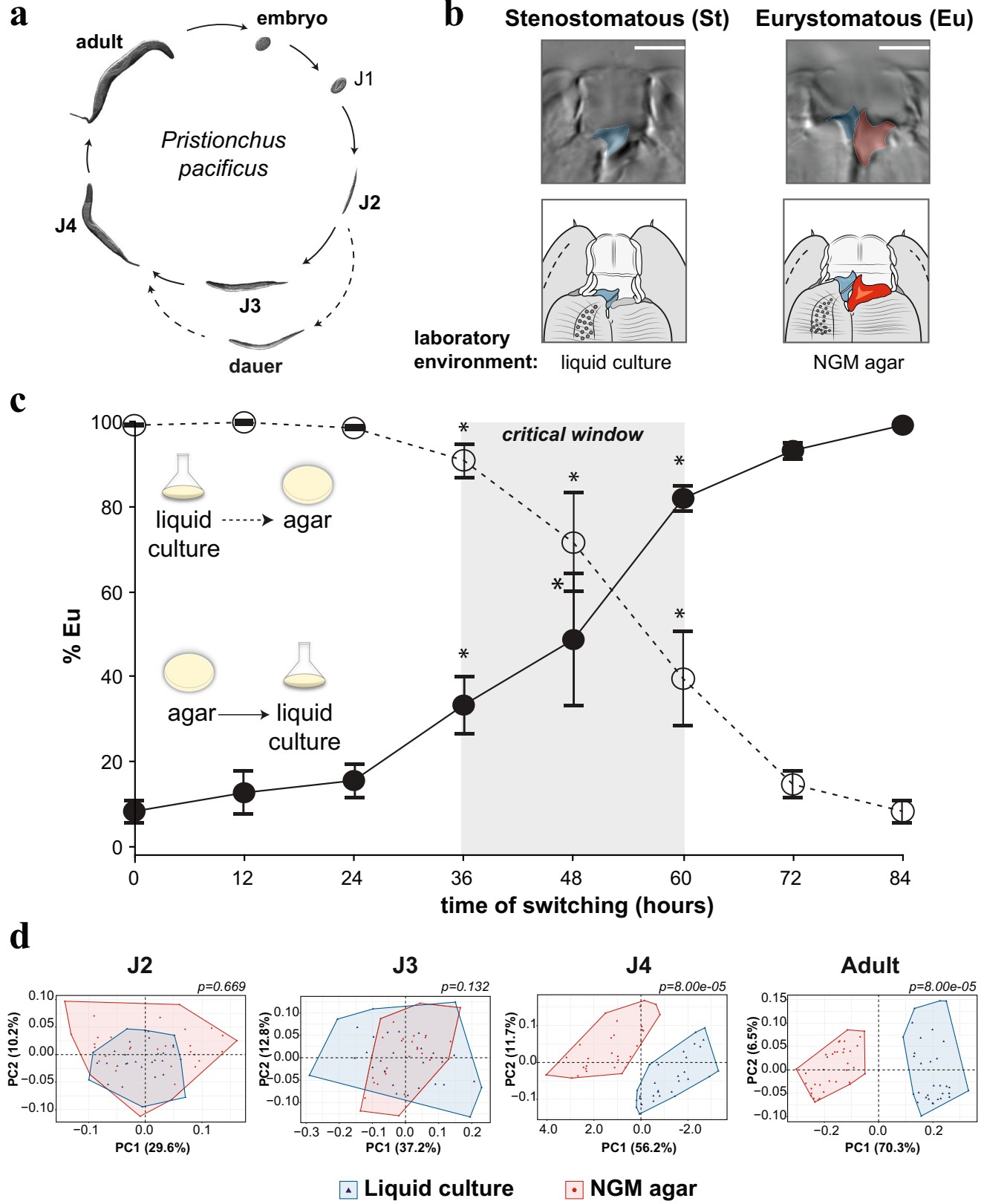

**b** Stenostomatous (St)    Eurystomatous (Eu)

laboratory environment:    liquid culture    NGM agar

**c** *critical window*

**d** J2 *p=0.669* | J3 *p=0.132* | J4 *p=8.00e-05* | Adult *p=8.00e-05*

▲ **Liquid culture**    • **NGM agar**

H4K5/12ac identified here also extends beyond gene regulation to an ecologically relevant developmental decision. Finally, we found that acetylation regulates developmental rate across multiple Ecdysozoan species, which may be linked to the establishment of the Eu morph in *P. pacificus*. Thus, our results reveal the molecular determinants of plasticity and their potential evolutionary origin.

## Results

### Reciprocal transplantation reveals a critical window of environmental influence

In the laboratory, different culture conditions can be harnessed to bias morph development: the wild type *P. pacificus* strain PS312 is predominantly St when grown in liquid S-Medium and Eu when grown on

**Fig. 1 | Mouth-form phenotypic plasticity is determined during a critical window in development. a** Life cycle of *Pristionchus pacificus*. **b** Stenostomatous (St, left) and Eurystomatous (Eu, right). Scale bar = 5 μm. **c** Switching experiments delineate developmental phases of plasticity. Adult phenotypes are plotted as a function of transferring between environments after synchronization. Statistical significance was assessed by binomial logistic regression on Eu and St counts between a given time-point, and the *t'* = 0- and 84-h phenotype; *p*-values were adjusted by Bonferroni correction, *$p < 0.025$ relative to both. Error bars represent S.E.M. for $n = 5$ independent worm populations. Note, intermediate values are intermediate ratios, not intermediate morphs. **d** GMM analysis of mouth-shape differences throughout development. PCA plots show the morphospace that contains shape variation per stage. Effect size (Z) for J2 = 1.403, J3 = 2.172652, J4 = 3.847113, Adult = 5.0949. *p*-values represent FDR-adjusted pairwise Procrustes Anova. Source data are provided as a Source Data file.

NGM agar plates[23]. We investigated the boundaries of environmental sensitivity by performing reciprocal transplant experiments between liquid and NGM-agar. Transferring between environments at 12 and 24 h after egg synchronization (t'0) led to a complete adoption of the second environment's phenotype, regardless of the direction of the environmental shift (Fig. 1c; $n = 5$, $p < 0.05$). Thus, this period represents a naïve and fully plastic phase of development. In contrast, when animals were switched at ≥36 h they began to retain memory of their previous environment. Specifically, an increasing percentage of animals executed the morph of the first environment (Fig. 1c). Finally, after 60 h, transplanting had little to no effect on mouth-form ratios. These results reveal a critical time window of environmental sensitivity between 36 and 60 h. Prior to that, juveniles are completely plastic while afterwards the decision is irreversible.

## Molecular and morphological plasticity underly the critical window

Next, we looked into potential mechanisms that provide plasticity during the critical window and end plasticity after it. First, we investigated the role of 'switch genes', which determine alternative phenotypes depending on their expression above or below a given threshold[24,25]. *eud-1* is a steroid sulfatase that yields 100% Eu animals when constitutively overexpressed and 100% St animals when knocked out[19]. We predicted that transcriptional plasticity of switch genes (i.e., *eud-1*) underlies phenotypic plasticity during the critical window, while invariant expression demarcates deterministic development. To test this prediction, we measured mRNA levels of *eud-1* during normal development and reciprocal transplant experiments. At 12 h, we found strong induction of *eud-1* in NGM-agar (Eu condition) compared to liquid culture (St condition)(Supplementary Fig. 1a, b, Supplementary Data 1). Surprisingly though, modest expression was still observed in liquid culture, indicating some amount of environment-insensitive transcription during the naïve phase. This modest expression is rapidly induced when worms are transferred to NGM-agar at 24 h, or decreased when transferred from NGM-agar to liquid (Supplementary Fig. 1a,b; $p < 0.05$), demonstrating environmental sensitivity. Intriguingly, *eud-1* begins to be repressed after 36 h regardless of the environmental condition. At 60 h *eud-1* mRNA levels are normalized between environments−coinciding with the end of the critical window.

Next, we wondered if plasticity during the critical window is confined to a specific juvenile stage/molt. First, we measured juvenile stages after hypochlorite-synchronization and compared these data to our reciprocal transplant data, which indicated that the critical window is centered in the J3 stage (Supplementary Fig. 1c–g). Second, fitting our reciprocal transplant data to a logistic model revealed inflection points between 48–54 h, which coincides with the J3-J4 molt (Supplementary Fig. 1h, i). Third, quantitative geometric morphometrics (GMM) throughout development revealed significant differences between conditions beginning in the J4 stage (Fig. 1d, $p < 0.05$ and $Z \geq 2.0$; Supplementary Fig. 2), even though mouth dimorphism was previously thought to occur only in adults[26]. Molecular and phenotypic divergence may differ depending on genotype and environmental conditions, and it's unclear if Eu J4s will necessarily become Eu adults after the final molt. Nevertheless, under these standardized conditions with the laboratory strain PS312, a pattern emerged: plasticity correlates with switch-gene transcriptional flexibility, while exit from the critical window correlates with switch-gene repression and morphological differentiation. We note that this pattern provides a plausible mechanism for the establishment of environmentally sensitive critical periods of development.

## Morph development is regulated by H4K5/12 acetylation

We hypothesized that open and closed chromatin structure−mediated by histone modifications[27-29]−underlies transcriptional plasticity *vs.* repression of the switch gene *eud-1*. First, we queried a panel of inhibitors that target histone-modifying enzymes for a potential effect on mouth form (Fig. 2a, b; see Methods for detailed description of statistics, $n \geq 3$). Interestingly, the Tip-60/KAT5 histone acetyltransferase inhibitor NU9056 converted normally Eu animals on NGM-agar to the St morph ($p = 1e$-15), while the histone deacetylase (HDAC) inhibitor Trichostatin A (TSA) converted normally St animals in liquid culture to the Eu morph ($p = 4.3e$-12). Hence, functionally opposite inhibitors yielded correspondingly opposite phenotypes, indicating that histone acetylation/deacetylation has an important role in mouth-form development.

NU9056-treated animals had pleiotropic effects (e.g., reduced fertility, egg laying) consistent with *Ppa-mbd-2* and *Ppa-lys-12* mutants that are egg laying or vulva defective and are St-biased[30]. However, animals that were exposed to TSA appeared wild type except for the effect on mouth form. HDAC inhibitors have also been shown to affect developmental decisions in other organisms, and several are currently in use as chemotherapeutic drugs[28,29,31,32]. Yet, how HDAC inhibition leads to these effects remains poorly understood. Therefore, in the following we focus on TSA and the potential role of histone acetylation/deacetylation in plasticity.

First, we confirmed that TSA also induced the Eu morph in axenic liquid culture, arguing for a direct inhibition of nematode enzymes rather than an indirect effect on their bacterial diet ($p = 1.2e$-6)(Supplementary Fig. 3a). TSA also increased the proportion of Eu animals in three different *Pristionchus* species (Supplementary Fig. 3b), indicating that acetylation has an evolutionary conserved role in mouth form across the genus. Surprisingly however, although TSA is a pan-deacetylase inhibitor[33], no obvious effect on mouth form was seen after treatment with several other HDAC inhibitors (Supplementary Fig. 3c). Presumably this is due to an unusual degree of specificity between TSA and its target enzyme in *Pristionchus*, and we wondered if that would allow us to investigate the role of discrete acetylated residues in plasticity.

To probe the potential specificity of TSA in *P. pacificus* we performed Western Blots (WB) on acid-extracted histones (Fig. 2c, Supplementary Fig. 4). In addition to DMSO (solvent) we used butyrate as a second negative control because it is an HDAC inhibitor that did not affect mouth form (Supplementary Fig. 3c). H3K27ac is a reproducible marker of active enhancers and promoters[27], and has recently been implicated in behavioral differences between ant castes[28,34]. However, we did not observe an increase in H3 acetylation using a pan-acetyl antibody or with a specific antibody toward H3K27ac ($n = 3$, Fig. 2d, e). In contrast, we observed a 2-fold induction of H4 acetylation (±0.1, $p = 6.3e$-4 *vs.* DMSO and 8.5e-3 *vs.* butyrate). To verify TSA's effect on H4 and to determine which H4 lysine(s) are hyperacetylated, we repeated WBs with specific H4-acetyl antibodies (Fig. 2f, g). Consistent with the apparent specificity of TSA in *Pristionchus*, we observed only

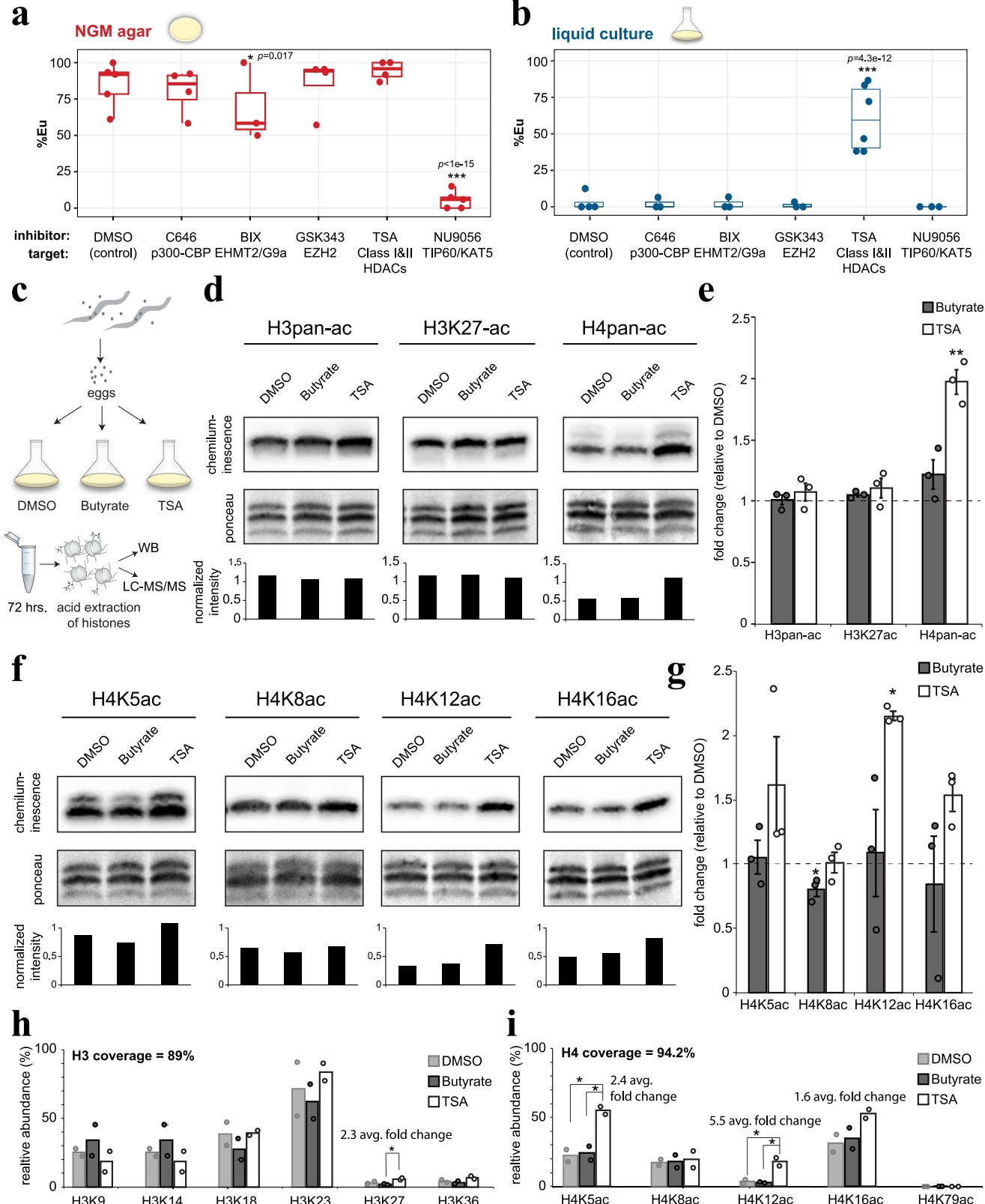

one H4 lysine with significant hyperacetylation relative to both controls: H4K12 (p=5.7e-6, 3.6e-2, respectively).

To further validate this result, we performed Liquid Chromatography Tandem-Mass Spectrometry (LC-MS/MS) (n = 2, Fig. 2c, and h,i, Supplementary Data 2). In agreement with our WB data, H4K12ac exhibited the greatest increase in TSA for all queried H3 and H4 lysine residues (avg. fold change = 5.5). The greater sensitivity of LC-MS/MS

revealed a smaller yet significant increase in H4K5ac against both DMSO and butyrate (avg. fold change = 2.4). No other modification was statistically significant against both controls (e.g., H3K27ac exhibited an increase relative to butyrate, but not to DMSO). Collectively, both immunostaining and mass spectrometry implicate H4 acetylation, and in particular H4K12ac, as the main effector by which TSA influences mouth-form development.

**Fig. 2 | Mouth form is linked to H4K5/12ac. a, b** Phenotype (%Eu) after exposure to histone-modifying enzyme inhibitors in each environment. Target enzymes indicated below inhibitors. Box-plot minima and maxima represent the 25 and 75% quantile, respectively, middle bar represents 50% quantile (median). Whiskers denote 1.5x the interquartile range. Statistical significance determined by binomial logistic regression on Eu and St counts, $n \geq 3$ biologically independent worm populations. *p*-values for agar-BIX = 0.017, agar-NU9056 = 1e-15, liquid-TSA = 4.3e-12. **c** Strategy to identify the molecular mechanism of TSA's effect: Bleach-synchronized eggs were seeded into liquid culture with DMSO (negative control for solvent), Butyrate (negative control for HDAC inhibitor that had no effect on phenotype), and TSA. Worm pellets were used for acid-extraction of histones and subject to WB and LC-MS/MS. **d** Representative WBs of *P. pacificus* histones with specified antibodies, total histone staining by ponceau, and resulting relative intensities by quantitative densitometry. **e** Mean fold change vs. DMSO, $n = 3$ biologically independent worm populations. Error bars represent S.E.M. Statistical significance was assessed by a two-tailed student's *t*-tests between TSA and Butyrate, *p*-value for H4pan-ac = 0.0085. **f, g** Same as **d**, **e** but with H4-specific antibodies, $n = 3$ biologically independent worm populations. *p*-value for H4K12ac = 0.036. **h** Relative H3 and **i** H4 acetylation of indicated residues compared to total H3 and H4 peptide intensities. Statistical significance was assessed by a 2-sided student's *t*-test, $n = 2$ biologically independent worm populations. *P*-values for H3K27ac: TSA vs. DMSO = 0.13, TSA vs. butyrate = 0.033. *P*-values for H4K5ac: TSA vs. DMSO = 0.025, TSA vs. butyrate = 0.033. *P*-values for H4K12ac: TSA vs. DMSO = 0.047, TSA vs. butyrate = 0.036.*$p < 0.05$, **$p < 0.01$. Source data are provided as a Source Data file.

## Genome-wide patterns of H4 acetylation and a role in switch-gene expression

The role of H4K12ac in development is not well understood, but there are links to aging[35], hormone-dependent cancer[36], and learning;[37] processes that are all connected to dynamic gene regulation. Previous Chromatin Immunoprecipitation and sequencing (ChIP-seq) experiments in human cells showed that H4K12ac is distributed more throughout gene bodies than other acetylated lysines, which primarily demarcate enhancers or promoters[27,38], hinting at a role for H4K12ac in transcription elongation rather than initiation. We investigated whether, and how, TSA effects switch-gene expression in *P. pacificus*. Using RT-qPCR, we found that TSA does increase *eud-1* expression compared to DMSO (Fig. 3a, Supplementary Fig. 3d, $p < 0.05$). However, instead of inducing *eud-1*, TSA appeared to prevent the repression that normally occurs after 36 h. These data suggest that H4K5/12ac does not have a role in environmental induction per se. Instead, hyperacetylation caused by TSA maintained the baseline of transcription beyond the critical window (36–48 h). These findings appear to differentiate H4K5/12ac from other histone acetylation marks that are thought to directly promote transcriptional initiation[39].

To explore this possibility further, and to examine comprehensive patterns of H4 acetylation, we performed ChIP-seq with antibodies targeting H4K5ac, H4K8ac, H4K12ac, and H4K16ac at 48 h in +/- TSA conditions. Consistent with our original biochemical results (Fig. 2), H4K12ac exhibited the greatest induction in TSA (Fig. 3b-e). H4K5ac exhibited the second largest induction, and correlated with H4K12ac abundance (Pearson correlations 0.82 and 0.84 per replicate; Supplementary Fig. 5a). H4K12ac peaks were significantly enriched in regulatory regions compared to genome-wide chromatin state distributions[40] ($p < 0.01$, Fisher's Exact Test; Supplementary Fig. 5b), and Gene Set Enrichment Analysis[41] of genes adjacent to hyperacetylated regions. (>1.5-fold change) revealed a significant association with 'positive regulation of programmed cell death' and an RNAi phenotype of 'cell proliferation increased' (Supplementary Fig. 5c–e; e.g. the putative vacuolar ATPase subunit *vha-10*). Programmed cell death is one of several competing models by which TSA and other HDAC inhibitors have been proposed to underly antitumor effects in humans[42–44]. Our results in *P. pacificus* suggest a deeply conserved role for acetylation in controlling pro-apoptotic gene expression and indicate that nematodes may be a useful model for investigating the chemotherapeutic mechanisms of HDAC inhibition.

## An H4 acetylation/deacetylation timer determines the critical window

Interestingly, although H4K12ac peaks were enriched in promoters and enhancers (Supplementary Fig. 5b), TSA led to H4K12 hyperacetylation primarily over gene bodies (Fig. 3d and Supplementary Fig. 5e). This pattern also appeared to be true at the locus encompassing the *eud-1* switch gene, which is part of a four-member 'supergene' on the X chromosome where each gene contributes to mouth form regulation[45]. Our ChIP-seq data revealed that the super-gene locus is bordered by large peaks of H4 acetylation (Fig. 3f, Supplementary Data 3), and that TSA led to an increase in H4K12ac between these peaks, including across *eud-1*, rather than at the peaks themselves (Fig. 3f, g).

Next, we performed ChIP-qPCR at 24 and 48 h to see if we could track deacetylation across *eud-1* during the critical window. Indeed, we found a significant decrease in H4K12ac at 3/3 qPCR primer locations spanning the *eud-1* gene locus (Fig. 3h–j, $p < 0.05$). However, adding TSA partially maintained the early juvenile levels of H4K12ac at 48 h (Fig. 3h–j, Supplementary Fig. 6c); arguing that TSA prevents *eud-1* repression by preventing H4 deacetylation. In contrast, we did not detect an increase in either H3 acetylation or H4K16ac at *eud-1* in the presence of TSA, confirming the specificity of our previous results (Fig. 3k). Interestingly, we also observed a broad distribution of the facultative heterochromatin mark H3K27me3 over the supergene locus flanked by the H4K12ac peaks (Supplementary Fig. 6a). However, in contrast to H4K12ac, we did not observe significant changes in H3K27me3 across development at *eud-1* (Supplementary Fig. 6b). Collectively, these data lead toward a model of early H4K5/12ac deposition in a facultative heterochromatin domain that permits switch-gene induction by the environment. During the J3-J4 transition, an unknown developmental mechanism leads to deacetylation, turning off switch-gene expression and effectively closing the critical window (Supplementary Fig. 7a).

To test this model, we first asked whether TSA requires *eud-1* to induce the Eu morph. Indeed, *eud-1* mutant animals remained 100% St even in the presence of TSA ($n = 3$, $p < 0.05$). Next, we wondered if the sulfatase inhibitor STX-64[46] can be used to inactivate EUD-1, which would allow temporal control of EUD-1 activity. Application of 1 μg/ml of STX-64 caused a 100% St phenotype on NGM-agar, mirroring the *eud-1* phenotype (Supplementary Fig. 7b). With this tool in hand, we grew a *eud-1* constitutive expression line (Ex[*eud-1*])[19] on STX-64 plates until 48 h, and then switched to DMSO plates. In theory, transferring off STX-64 plates should remove inhibition, and thereby test whether EUD-1 activity after 48 h is sufficient to induce the Eu morph. Unswitched control Ex[*eud-1*] worms exhibited a 49% Eu phenotype, yet switched Ex[*eud-1*] worms exhibited a 98% Eu phenotype (Supplementary Fig. 7c; $n = 3$, $p < 0.05$). Thus, EUD-1 activity after the critical window is sufficient to induce the Eu morph—consistent with our model that prolonged *eud-1* expression by hyperacetylation induced the Eu morph in our TSA experiments.

## Epigenetic memory can be enforced by HDAC inhibition

The role of histone modifications in plasticity provides the potential for epigenetic gene regulation; a long-held but poorly supported hypothesis to connect plasticity to evolution[5]. To determine whether histone acetylation can provide long-term memory, we assessed if preventing deacetylation during the critical window would 'fix' or 'freeze' an initial developmental trajectory despite shifting to a different environment. In principle, this would provide compelling evidence that histone modifications can carry long-term environmental

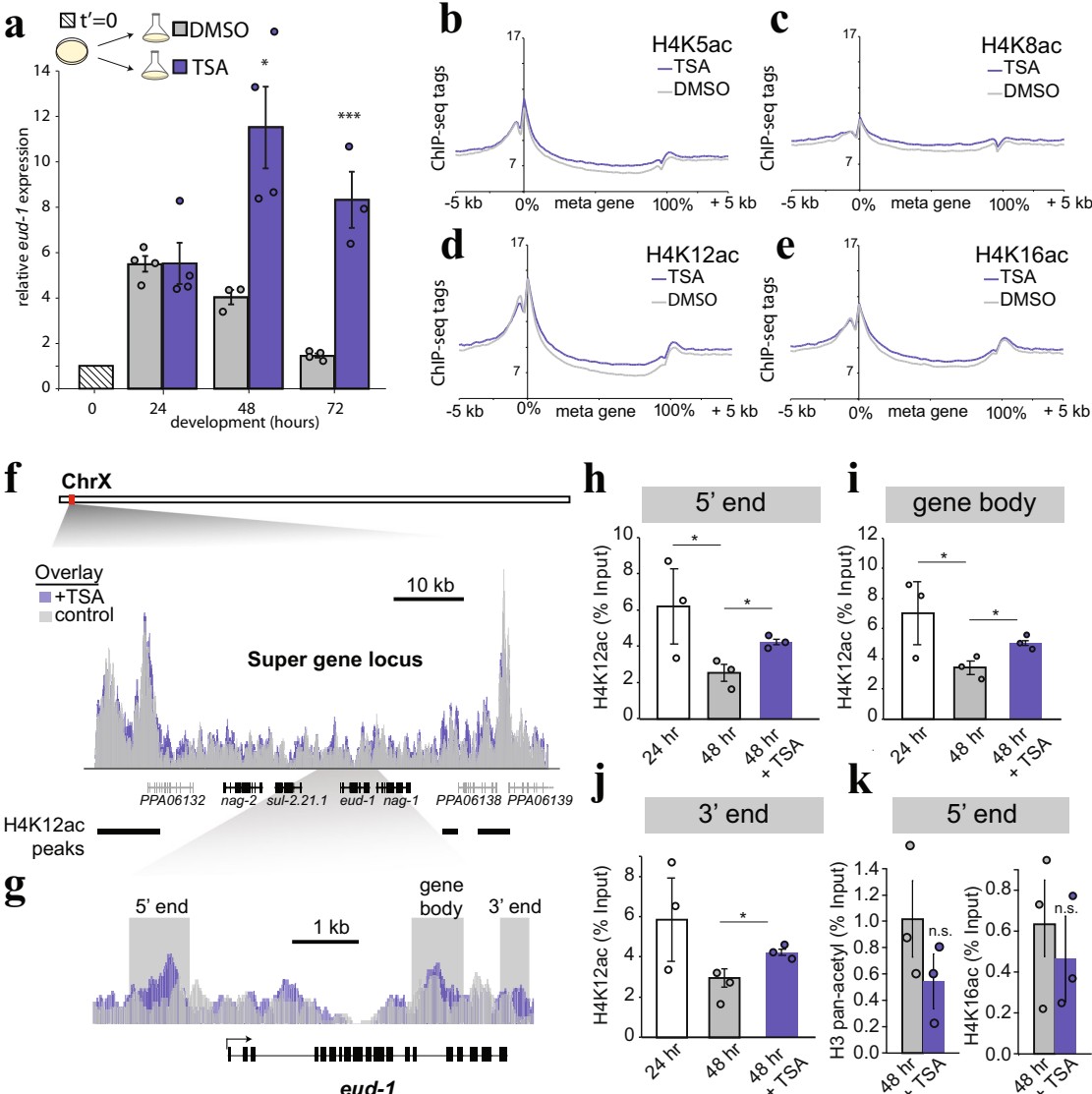

**Fig. 3 | H4K12ac acetylation/deacetylation at switch genes determines plasticity and canalization, respectively. a**, *eud-1* expression at indicated time points by RT-qPCR with either DMSO or TSA added directly after bleach synchronization. Error bars represent S.E.M. of 4 biologically independent replicates except for DMSO 48 h and TSA 72 h where *n* = 3. Statistical significance was determined by student's *t*-test. *P*-values for 48-h comparisons = 0.0089, and 0.00062 for 72-h comparisons; *\*p* < 0.05, *\*\*p* < 0.01, *\*\*\*p* < 0.001. **b**−**e** ChIP-seq meta-gene profiles of H4-acetylated residues at 48 h in +/- TSA. y-axis represents average input-normalized ChIP-seq tag counts per bp per gene, and the *x*-axis represents 5' to 3' gene coordinates divided into 200 bins +/− 5 kb, *n* = 2. **f** ChIP-seq tag density of H4K12ac over the multi-gene locus (black genes) that control mouth-form plasticity, which are flanked by large peaks. y-axis = 0-72 depth-normalized density for both tracks. Black bands indicate H4K12ac peaks in liquid culture determined by MACS2. **g** Zoom in on *eud-1* and regions targeted by qPCR in **h**−**k**. **h**−**j**, ChIP-qPCR of H4K12ac across *eud-1* at 24 h, 48 h and 48 h when grown in TSA, *n* = 3 biologically independent worm populations. **k** ChIP-qPCR of H3-pan-acetyl and H4K16ac at the 5' end of *eud-1*, *n* = 3 biologically independent worm populations. Statistical significance in '**h**−**k**' was assessed by a 1-tailed student's *t*-test. *P*-values for '**h**': 24-h vs. 48-h = 0.043, 48-h vs. 48-h+TSA = 0.013. *P*-values for '**i**': 24-h vs. 48-h = 0.042, 48-h vs. 48-hr+TSA = 0.016. *P*-values for '**j**': 48 h vs. 48 h + TSA = 0.017. *\*p* < 0.05 Source data are provided as a Source Data file.

information to affect future developmental decisions. To test this premise, we combined transplant experiments with TSA treatment. Specifically, we transferred worms between NGM-agar and liquid at 24 h, which normally leads to the St phenotype (Fig. 1c), but this time added TSA at the time of switching. Remarkably, these worms were phenotypically similar to having experienced the agar environment for their entire development (90.8% Eu ±1.2)(Fig. 4). Not only were environmentally shifted TSA-treated worms significantly different from DMSO controls, but they were also significantly different from worms treated with TSA at *t'* = 0 without switching (60.9% Eu ±8.0). These results demonstrate a combined effect of the environment with HDAC inhibition that is consistent with fixing the initial Eu developmental trajectory. We then performed a series of experiments to attempt to

refute this interpretation (Fig. 4). First, to rule out that simply adding TSA at a later time point has a greater effect on mouth form than at *t'* = 0, perhaps due to degradation, we added TSA at 24 or 48 h without environmental transfer. These experiments resulted in decreasingly intermediate ratios of Eu animals similar to, and less than adding TSA at *t'* = 0, respectively (*p* < 0.05). We also repeated transplant experiment at 24 h, but waited until 48 h to add TSA. Again, this experiment failed to induce a typical NGM-agar phenotypic ratio (28.5% Eu ±3.8). These results show that there is an additive effect of the environment with TSA, and that this effect depends on adding TSA at the precise time of environmental transfer. We conclude that preventing histone deacetylation after a temporary juvenile experience in NGM-agar fixed the Eu developmental trajectory; 48 h and two molts prior to adult

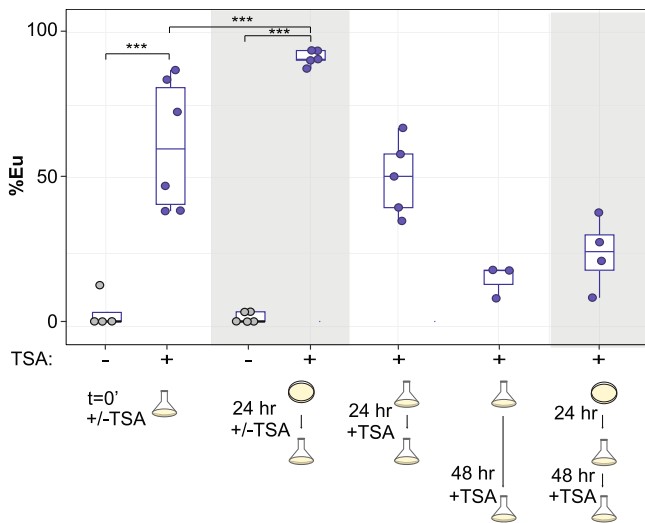

**Fig. 4 | H4 acetylation can provide long term memory.** Transplant experiments between NGM-agar and liquid culture +/- TSA at different time points. When TSA is not added (-), the solvent DMSO was added. The first two experimental conditions (DMSO and TSA) are from Fig. 2a. Individual data points are shown for all conditions; $n = 4$ independent worm populations x DMSO, 6 x TSA, 5 × 24 h switch +/-TSA, 5 × 24 h TSA, 3 × 48 h TSA, 4 × 24 h switch + 48 h TSA. Box-plot minima and maxima represent the 25 and 75% quantile, respectively, middle bar represents 50% quantile (median). Whiskers denote 1.5x the interquartile range. Statistical significance determined by binomial logistic regression on Eu and St counts. *P*-value for TSA vs. DMSO = 8.69e-13, <2e-16 for TSA vs. DMSO 24 h switch, and 7.67e-09 for TSA vs. TSA 24 h switch, *$p < 0.05$, **$p < 0.01$, ***$p < 0.001$. Source data are provided as a Source Data file.

differentiation. Thus, memory of environmental exposure can be stored and erased by acetylation and deacetylation, respectively.

### TSA treatment delays development in worms and flies

The extended *eud-1* expression in TSA was reminiscent of repeated or delayed developmental programs seen in some heterochronic gene mutants[47]. It was also previously shown that the Eu morph develops ~6 h slower than the St morph[48]. Therefore, we speculated that acetylation/deacetylation might also affect plasticity by affecting developmental speed. We coupled hypochlorite-treatment with temporary starvation to obtain highly synchronized cultures, and compared developmental rates in the presence or absence of HDAC inhibition. With this higher-resolution staging we found that TSA prolonged development by several hours (Fig. 5a, $p < 0.05$, two-way ANOVA). Most stressful or toxic treatments that delay development (i.e., excess DMSO, ethanol, low temperatures[49]) lead to St phenotypes, arguing for a more specific relationship between acetylation, development, and the Eu morph. To explore the generality of this phenomenon, we repeated these experiments in *Caenorhabditis elegans*, and again found a similar developmental delay (Fig. 5b, $p < 0.05$, two-way ANOVA). Importantly, *C. elegans* lacks mouth-form plasticity, implying that the effect on development is ancestral to the effect on mouth form. Recent data suggest that the divergence time between *C. elegans* and *P. pacificus* is 100-200 million years ago[50], hinting at a deeply conserved mechanism to control development timing. Indeed, when searching the literature, we found a similar effect of TSA on development had been reported in *Drosophila melanogaster* in 2001[51], which we independently confirmed (Fig. 5c, $p < 0.05$, two-way ANOVA). Thus, HDAC inhibition delays development in three highly diverged species of Ecdysozoa, a superphylum of molting animals spanning hundreds of millions of years[50].

### Histone acetylation may regulate plasticity by licensing developmental speed

Finally, we examined whether H4 hyperacetylation is the cause of delayed development. We found a 2.2-fold induction of H4K12ac in the presence of TSA in *C. elegans*, and a 2.6-fold induction in *D. melanogaster* (Fig. 5d–g, $p < 0.05$), in good agreement with our results from *P. pacificus*. However, we also saw an increase in H3 acetylation in *C. elegans* and *D. melanogaster*, consistent with previous reports that TSA has pan-deacetylase activity[33]. Given that both mouth form and developmental timing in *P. pacificus* are regulated by deacetylation, but control of timing appears to be ancestral, we speculate that developmental licensing was co-opted during the evolution of mouth-form plasticity. In this model, slowing down development would enable a longer period of switch-gene expression prior to the critical window.

## Discussion

Developmental plasticity allows organisms to adjust development to match their environment[5]. Environmental sensitivity is often limited to a specific stage of development, referred to as the critical 'window' or 'period'. Here, we show that entry and exit of the critical window determining *Pristionchus* mouth form is defined by H4K5/12 acetylation and de-acetylation. Moreover, we demonstrate that acetylation in juveniles can determine adult phenotypes.

HDAC inhibition also affects ant caste behavior[28,52], beetle horn size[29] and ocular dominance in mammals[53], suggesting that acetylation may be a common mechanism of regulating developmental plasticity. Yet, identifying which specific modification(s) control plasticity has been complicated by the number of acetyllysine sites on both histone 3 and 4. The role of H4K5/12ac in mouth form was uncovered in part due to our unbiased biochemical approach and in part to the unusual specificity of TSA in *P. pacificus*. H4 N-terminal tail acetylation is broadly correlated with gene activation[54], however, the functions of most individual acetyllysines are not well known. H4K16ac is the exception, and has been shown to be necessary for dosage compensation in flies[54] and hematopoietic differentiation in mammals[55]. However, we did not observe a significant correlation of this mark with switch gene transcription, or with commonly studied enhancer/promoter modifications on H3[27]. There is indirect evidence though for a role of H4K12ac in plasticity. For instance, H4K12ac has been linked to acetyl-CoA levels[35], suggesting that it may connect diet or metabolism to changes in gene expression. Furthermore, stimulating H4K12ac can promote memory formation, a paradigm of neuronal plasticity[37,56]. While the outcomes of neuronal and morphological plasticity are clearly different, the proximate mechanisms regulating plasticity may be shared.

Our time-resolved data also support a mechanistic role for H4K5/12ac in transcriptional elongation. This has been hinted at by ChIP-seq data showing an abundance of H4K12ac in gene bodies compared to other acetylated lysines[38,57]. Furthermore, the bromodomain and extraterminal domain (BET) family protein BRD4, which binds to acetylated histones, was recently shown to act as an elongation factor to facilitate RNAPII clearance through chromatin[58]. BRD4 promotes estrogen receptor-positive breast cancer by interacting with hyperacetylated H4K12[36], and the related protein BRD2 interacts specifically with H4K5/12ac in immortalized human cell lines[36,59,60]. It may be worth investigating whether H4K5/12ac has conserved roles in hormone-dependent processes by licensing transcription elongation at key switch genes[61,62]. Alternatively, TSA has also been shown to induce cell cycle arrest[63], which could explain its effect on development. Untangling these mechanistic possibilities may provide insight into both fundamental developmental biology and drug design.

The identified role of histone acetylation in plasticity also provides a potential avenue for epigenetic information storage. However, the half-life of histone acetylation is generally considered too short

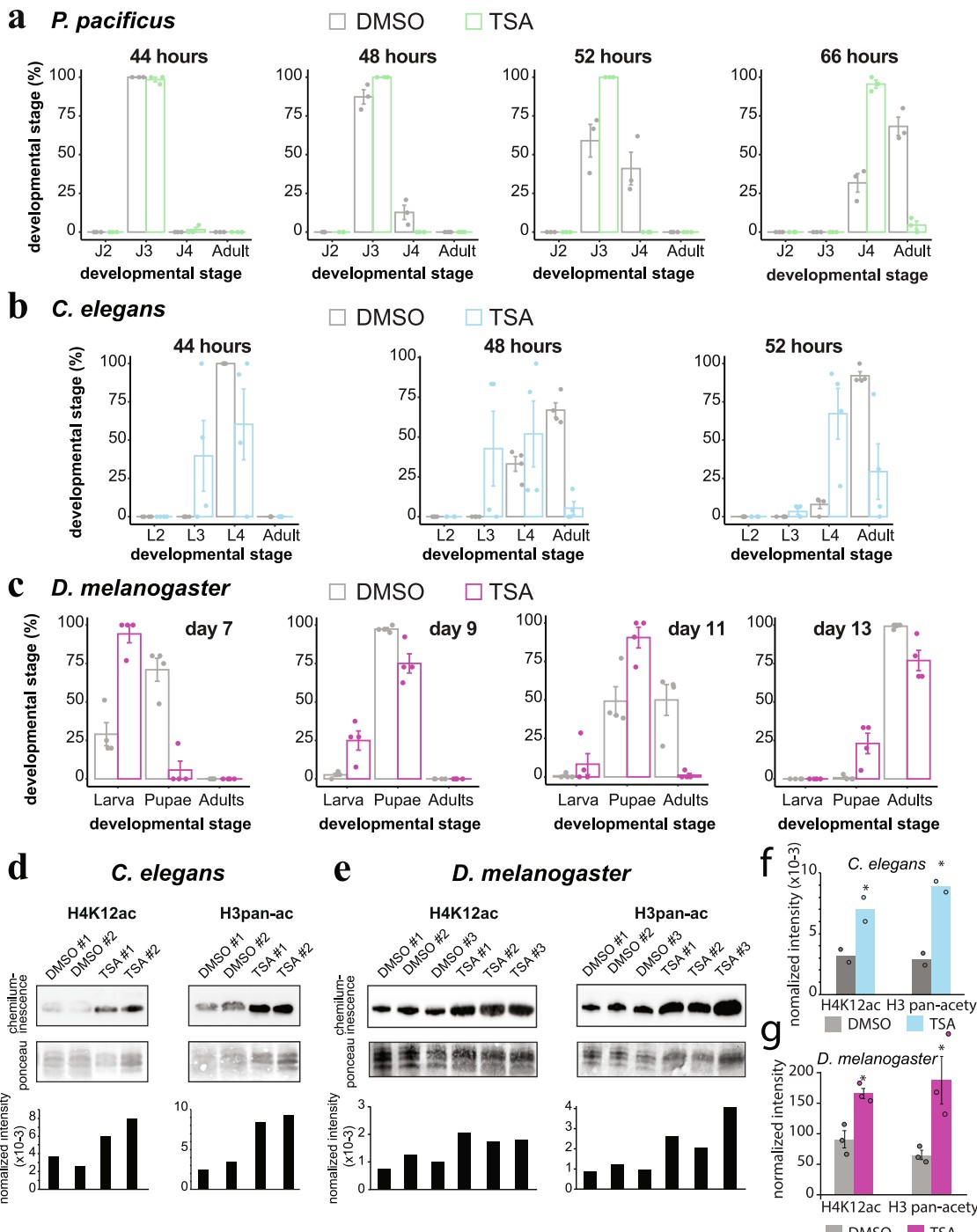

**Fig. 5 | Histone acetylation links plasticity to developmental speed.**
**a** Developmental rate of *P. pacificus*, **b** *C. elegans*, and **c**, *D. melanogaster* +/- TSA, $n = 3$ biologically independent populations of *P. pacificus*, 4 for *C. elegans*, and 4 for *D. melanogaster*. Significance was assessed with a two-way ANOVA. Error bars represent S.E.M. **d** Representative WBs of acid-extracted histones from *C. elegans* and **e** *D. melanogaster*. Intensity of chemiluminescence was normalized to total histone levels detected by Ponceau S. **f** Mean normalized intensity of WBs in *C. elegans* ($n = 2$ biologically independent populations) and **g** *D. melanogaster* ($n = 3$ biologically independent populations). Statistical significance determined by 1-tailed students *t*-test. *P*-values for *C. elegans* H3pan-ac = 0.0061, H4K12ac = 0.038. *P*-values for *D. melanogaster* H3pan-ac = 0.019, H4K12ac = 0.0052. Error bars represent S.E.M. Source data are provided as a Source Data file.

(1–2 h) to provide long-term memory[64]. Nevertheless, more recent studies have shown that histone acetylation can persist through mitotic division;[59,65,66] reviving the question of their epigenetic potential. Our results add an important next layer by demonstrating that histone acetylation can provide long-term environmental memory in a multicellular organism. More broadly, our results also provide empirical and mechanistic evidence that developmental plasticity is channeled through histone modifications.

Finally, the confluence of developmental timing and plasticity has intriguing implications for the evolution of novel phenotypes. Regulating developmental timing, referred to as heterochrony, has been hypothesized for nearly a century to facilitate evolutionary novelty[67]. While ample morphological evidence supports this hypothesis, there are scant known molecular mechanisms. Our results suggest that acetylation/de-acetylation acts as a timer to progress through development, and as a lever to regulate switch gene (*eud-1*) expression. In

*Pristionchus*, prolonging deacetylation could delay development and extend switch gene expression into the critical window, which would initiate the Eu morph. If sufficiently selected upon, the Eu morph would ultimately canalize, as appears to have occurred in some *Pristionchus* species[68], potentially representing plasticity-first evolution[69]. Going forward, it will be important to evaluate how H4K5/12 acetylation affects both molecular and evolutionary mechanisms.

## Methods

### Environmental switching experiments

Gravid adult *P. pacificus* were bleach-synchronized[70] and eggs were seeded into agar or liquid media as described in Werner et al. 2017[23], and then switched to the other condition at the indicated time points. At 84 h post-synchronization, adult worms were phenotyped with a 100x/1.4 oil DIC objective (Zeiss), and percent Eu was plotted on the y-axis. Statistical significance was assessed by binomial logistic regression on Eu and St counts between a given time point and both the t'0 and the 84 h phenotype (see section below on 'Statistics on mouth-form frequencies' for more details). Error bars represent S.E.M. for n=5 independent biological replicates, with >20 worms counted per replicate.

### Statistics on mouth-form frequencies

To test the statistical significance of the effect of small molecule inhibitors or switching experiments on mouth-form frequencies, while incorporating biological replicates, we performed a weighted binomial logistic regression in the R base statistical software package. Eu counts were modeled as 'successes' and St modeled as 'failures,' and replicates were included as repeated measurements: summary(glm(formula = cbind(Eu,St)~condition, data = results_table.txt, family = "binomial"))

Resulting *p*-values were adjusted for multiple testing using a *bonferroni*-correction (i.e., by the number of inhibitors tested), and are indicated in corresponding figure panels. $n \geq 3$ in all cases, with individual replicates shown as data points.

### Fitting reciprocal transplant data

Raw data of transplantation experiments were normalized on the y-axes to '% plasticity' by subtracting the average minimum value (e.g., the average percent Eu when grown for the entire duration of development in liquid) from each time point. The normalized data were fit to a four-parameter logistic function, which is commonly used in pharmacology to fit a physiological response to drug concentration[71] or ligand binding to a protein in biochemistry (i.e., the Hill equation):

$$\%\text{plasticity} = a + \frac{(b-a)}{1 + \left(\frac{K_T}{\text{time point}}\right)^{-n}}$$

Where $a$ = the minimum asymptote (or plasticity after determination), $b$ = maximum asymptote (or plasticity at $t' = 0$), $K_T$ = the inflection point of the curve (or the time point when plasticity transitions to determination, equivalent to the effective concentration $EC_{50}$ in pharmacological studies, or $K_A$ in protein-binding experiments), and $n$ = the slope at the steepest part of the curve (or the rate of determination, also known as the Hill coefficient). Note, $K_T$ is not necessarily equivalent to 50% plasticity because the maximum asymptote or maximum plasticity $b$ is not necessarily equal to 100%. To fit the data, a nonlinear least squares regression was performed in R using the 'nls' function with upper bound = 100, lower bound = 0, and starting estimates of $a = 0$, $b = 100$, $K_T = 60$, and $n = 10$, which reached convergence after four iterations. The fitted values were plotted using 'ggplot2,' and a line marking the inflection point $K_T$ was added by 'geom_vline.'

### Geometric morphometric (GMM) quantification

Two-dimensional mouth shapes were quantified using 13 fixed landmarks and 3 sliding semilandmarks. Landmark configurations from $n \geq 30$ animals per stage and condition were used as input for shape analysis by General Procrustes Alignment (GPA), and PCA was performed on the aligned Procrustes coordinates (i.e., shapes) as described in Theska et al.[72]. GPA was performed using 'gpagen' of geomorph (ver. 4.0.0); sliding of the semilandmarks was achieved by minimizing bending energy. Statistical testing for differences between group means of shape variation was performed on Procrustes coordinates of animals of the same stage in different conditions via permutational Multivariate Analysis of Variance (*PERMANOVA, or* 'Procrustes' ANOVA) with a randomized residual permutation procedure (RRPP). This was done using the 'procD.lm' function of geomorph with 100,000 iterations, seed = NULL. Resulting relative effect sizes (*Z*-scores) and FDR-adjusted *p*-values are presented adjacent to corresponding PCA plots. We considered a PERMANOVA effect as 'incompatible with the null hypothesis' if the relative effect size was greater than or equal to two times the standard deviation (i.e., $Z \geq 2.0$) and the associated *p*-value was below a type I error rate of 0.05 (i.e., $p < 0.05$).

### Histone-modifying-enzyme inhibitor assays

Worm cultures were synchronized by adding bleach solution (1 ml bleach/0.5 ml 5M NaOH) to 3.5 ml of gravid adult worms, which were washed with M9 from 3 x 6 cm NGM agar plates per experimental condition. Eggs were then aliquoted to 10 ml S-medium liquid cultures or 6 cm NGM agar plates. For liquid experiments, inhibitors were added directly at the time of aliquoting eggs. For agar experiments, fresh 6 cm NGM plates were spotted with 500 µl of overnight cultures of *E. coli* OP50 (37 °C in LB, 180 rpm) plus inhibitors and spread evenly over the surface of the plate, then air-dried in a chemical hood before adding 100-500 bleached eggs. The following concentrations of inhibitors or control (DMSO) were added: 100 µl DMSO (Sigma Aldrich cat. #D8418) = 1% (v/v), 100 µl 6.6 mM TSA (Selleckchem cat. #S1045) dissolved in DMSO = 66 µM final conc., 100 µl 2.2 mM C646 (Sigma Aldrich cat. # 382113) dissolved in DMSO = 22 µM final conc., 100 µl 1.7 mM BIX (Sigma Aldrich cat. #B9311) dissolved in DMSO = 17 µM final conc., 100 µl 1.85 mM GSK343 (Sigma Aldrich cat. #SML0766) dissolved in DMSO = 18.5 µM final conc., and 30 µl 43 mM NU9056 (Tocris cat. #4903) dissolved in DMSO = 129 µM final conc.. For Fig. S2, inhibitors were resuspended at the following concentrations before titrating into LC: 1 M Na-butyrate (Sigma Aldrich cat. #B5887) dissolved in water, 1 mM apicidin (Sigma Aldrich cat. #A8851) dissolved in DMSO, 10 mM CI-944 (Sigma Aldrich cat. #C0621) dissolved in DMSO, 10 mM TMP195 (Selleckchem cat. #S8502) dissolved in DMSO, 10 mM Pyroxamide (Sigma Aldrich cat. #SML0296) dissolved in DMSO, 5 mM SAHA/Vorinostat (Sigma Aldrich cat. #SML0061) dissolved in DMSO, 4 mM Tubacin (Sigma Aldrich cat. #SML0065) dissolved in DMSO, 10 mM CUDC-101 (Selleckchem cat. #S1194) dissolved in DMSO, 10 mM Panobinostat (MedChemExpress cat. #HY-10224) dissolved in DMSO, and 10 mM Belinostat/PXD101 (MedChemExpress cat. #HY-10225) dissolved in DMSO.

### Histone purification and western blot

For nematodes, crude nuclei were obtained as in Werner et al. 2018[40] but without sucrose cushion purification, with starting inputs of 200–500 µl worm pellets (10–20 × 10 cm plates of bleach-synchronized worms) collected at 72 h in the presence of 100 µl DMSO, 66 µM TSA, or 10 mM butyrate (see 'Histone-modifying-enzyme inhibitor assays' method section for resuspension and cat. #s). Histones were acid-extracted from crude nuclei, and precipitated in TCA (Sigma Aldrich cat. #T9159) following Shechter et al.[73], and resuspended in 80 µl water. Histone yields from each purification were determined by using a calibration curve of recombinant H3 and H4 (NEB, cat. #M2503S, #M2504S, respectively). Absolute amounts of histone were determined by densitometry on coomassie-stained bands with Fiji[74] for *Pristionchus* data (conducted at the MPI in Tübingen) and Image Lab software (BioRad) for *C. elegans* and *D.*

*melanogaster* (conducted at the University of Utah). For Western Blot, 5 μg of histone sample was loaded per lane on a BioRad 'any Kd' Precast gel (cat. #4569033) alongside 5 μl PageRuler Prestained Protein Ladder (Thermo Scientific cat. #26616), and run 200V for 28 min. A Wet Transfer was performed with Bjerrum Schaffer-Nielsen Buffer + SDS (48 mM Tris, 39 mM glycine, 20% methanol, 0.0375% SDS, pH 9.2) in a Mini Protean Tetra box (BioRad) to 0.2 μM, 7 × 8.5 cm pre-cut nitro-cellulose membranes (BioRad, cat. #162-0146), at 4 °C with magnetic stirring and an opposing cold-pack, 100V for 10 min, then 60V for 20 min. Total-protein transferred was visualized by 10-20 ml 0.1% ponceau/5% acetic acid for 5–10 min with rotation, washed with dis-tilled water until bands became apparent, and then imaged on a Quantum gel imager (Vilber Lourmat) with white light using the 'pre-view' function (in Tübingen) or BioRad ChemiDoc (Utah). Nitrocellu-lose blots were then briefly washed in TBS (50 mM Tris-HCl, 150 mM NaCl, pH 7.5), then blocked with 20 ml 5% nonfat dry milk (BioRad, cat #170-6404) in TBS for 1 h with rotation. Membranes were then washed 2 x 5–10 min in TBS plus 0.05% Tween 20 (TBS-T). Primary antibodies (Supplementary Data 1) were then incubated at 1:1000–2000 dilution in 5 ml 2.5% nonfat dry milk/TBS-T with membranes overnight (-12 h) at 4 °C with rotation. The next day, membranes were washed 4 x with TBS-T, 5 min each with rotation. Secondary antibodies corresponding to the animal immunoglobulin of the primary antibody, fused to horseradish peroxidase (Anti-rabbit IgG-HRP, Cell Signaling, cat. #7074S), were then incubated at 1:2,000 in 5 ml 2.5% nonfat dry milk/TBS-T for 1 h at room temperature. Membranes were then washed 4 x with TBS-T, plus one additional wash in TBS to remove residual Tween 20. To image, membranes were incubated in a 5 ml 1:1 A:B solution of Clarity Western ECL substrate (BioRad cat. #170-5060) for 5 min at room temperature with rotation, and chemiluminescence was detec-ted on a Fusion SL imager (Vilber Lourmat; Tübingen) or ChemiDoc (BioRad; Utah) within 5 min.

For *Drosophila* experiments, flies were cultured on standard medium containing cornmeal, yeast, agar, and molasses, and main-tained at 25 °C and 60% humidity on a 16:8 light:dark cycle. Addition-ally, the medium contained either TSA dissolved in DMSO, or DMSO only as a control. These were added to the medium at a concentration of 0.34% (v/v) once it had cooled to ~50 °C. This resulted in a final concentration of 10 μM TSA in the TSA-containing medium. The gen-otype of all flies used was *w1118* (Bloomington Drosophila Stock Center #3605). For crude histone purification, between 100 and 150 wander-ing third-instar larvae were collected per replicate, and three replicates per treatment were used. Larvae were rinsed twice in PBS, and histone extraction, purification, and quantification were performed as descri-bed above. Western blots were performed as described above, except only 1 μg of histone sample was loaded per lane, and images were acquired using a ChemiDoc MP Imaging System (BioRad). Image and statistical analysis were performed as described above, except two-tailed F tests were conducted to ensure that variances did not differ significantly between treatments, and one-tailed Student's *t*-tests were used since evidence from nematodes indicated that TSA was expected to cause increased histone acetylation. Antibodies used: H3pan = Active Motif #39140, lot #34519009 (1:2,000); H4pan = Active Motif #39926, lot #18619005 (1:1,000); H4K12 = Millipore #04-119-S, lot #3766681 (1:1,000). Intensities of chemiluminescent bands were quantified and normalized to intensities of histone bands from Pon-ceau staining. Statistical testing on normalized intensities was per-formed by a two-tailed *student's t*-test in nematodes and one-tailed test in *Drosophila*, $n = 3$ independent biological replicates for all experiments.

## Histone Liquid Chromatography Tandem-Mass Spectrometry (LC-MS/MS)

Histones were extracted from 72-h worm pellets as described for Western Blots (see above) for two independent biological replicates

each of 1% DMSO, 10 mM Na-Butyrate, and 66 μM TSA. 50–100 μg of soluble histones were then reduced with 1 mM DTT/50 mM ammonium bicarbonate for 1 h at room temperature, and then alkylated for 1 h in the dark with 10 mM chloroacetamide/50 mM ammonium bicarbo-nate. The reaction was neutralized with 10 mM DTT for 30 min. A 10x Arg-C digestion buffer was then added to histones (1x = 5 mM CaCl2, 0.2 mM EDTA, 5 mM DTT), which were digested with 1:50 Arg-C pro-tease (Promega, cat. #V1881): histone protein at 37 °C for 12–16 h. Digest completion was assessed by running an analytical SDS-PAGE of digested sample with undigested control sample. When complete, the reaction was stopped by adding 10% trifluoroacetic acid to a final concentration of 0.5%. Digested histones were then purified on homemade desalting C18 stage-tips[75], and run on an Easy-nLC 1200 system coupled to a QExactive HF-X mass spectrometer (both Thermo Fisher Scientific) in three technical replicates per biological sample as described elsewhere[76] with slight modifications: peptides were separated with a 127-min segmented gradient from to 10-33-50-90% of HPLC solvent B (80% acetonitrile in 0.1% formic acid) in HPLC solvent A (0.1% formic acid) at a flow rate of 200 nl/min. The 7 most intense precursor ions were sequentially fragmented in each scan cycle using higher energy collisional dissociation (HCD) fragmentation. In all measurements, sequenced precursor masses were excluded from further selection for 30 s. The target values for MS/MS fragmentation were $10^5$ charges and $3 \times 10^6$ charges for the MS scan.

Data was analyzed by MaxQuant software version 1.5.2.8[77] with integrated Andromeda search engine[78]. Acetylation at lysine was spe-cified as variable modification, and carbamidomethylation on cysteine was set as fixed modification. Endoprotease ArgC was defined as pro-tease with a maximum of two missed cleavages and the minimum peptide length was set to five. Data was mapped to the 'El Paco' protein annotation version 1 and 286 commonly observed contaminants. Initial maximum allowed mass tolerance was set to 4.5 parts per million (ppm) for precursor ions and 20 ppm for fragment ions. Peptide, protein and modification site identifications were reported at a false discovery rate (FDR) of 0.01, estimated by the target/decoy approach[79]. The label-free algorithm was enabled, as was the "match between runs" option[80]. A spectrum quality control threshold score >100 and posterior error probability (PEP) <0.01 was defined. The averages of two biological replicates of acetylated-peptide intensities normalized to total H3 and H4 peptide intensities are presented in Fig. 2h, i, and statistical significance was assessed by a two-way *stu-dent's t*-test. For fold-change, relative ion intensities of TSA-treated samples were compared to both DMSO and Na-Butyrate.

## Relative *eud-1* expression by Reverse Transcription-quantitative PCR (RT-qPCR)

Worm pellets (25–100 μl) collected at the indicated time points from each culture condition were freeze-thawed 3x between liquid nitrogen and a 37 °C heat block, then re-suspended in 500 μl Trizole (Ambion cat. #15596026). RNA was extracted following the manufacturer's protocol, then purified using a Zymo RNA clean & concentrator-25 columns. RNA was eluted in 50 μl water and quantified by nanodrop (A260/280). 1 μg RNA was used for reverse transcription with Super-Script II (Thermo Fisher cat. #18064071) following manufacturers protocols. Template RNA was degraded by the addition of 40 μl base (150 mM KOH, 20 mM Tris) for 10 min, 99 °C. cDNA pH was subse-quently neutralized by 40 μl acid (150 mM HCl), then diluted by 100 μl TE (200 μl final volume, 1:100 dilution). Four μl cDNA was used as input for 10 μl qPCR reactions with 2x Luna Universal qPCR Master Mix (NEB cat. #M3003X) and 0.25 μM forward and reverse primers (Supple-mentary Data 1) on a LightCycler 480/II 384 (Roche, serial #6073). Four technical replicates were performed per biological replicate qPCR primer set. Standard quantitative PCR thermocycling conditions were used: 95 °C for 5 min to denature, followed by 45 cycles of 95 °C for 10 seconds (s) (4.8 °C/s), 60 °C for 10s (2.5 °C/s), and 72 °C for 10s

(4.8 °C/s). After qPCR, a thermal melting profile was also obtained for each primer set and verified for whether it was a single peak.

For all experiments quantifying *eud-1* expression by RT-qPCR, threshold Ct values were compared to housekeeping genes *Ppa-cdc-42* and *Ppa-Y45F10D.4*[81] to obtain $2^{\Delta Ct}$, and the geometric mean was calculated for each time point, then normalized to *t*'0 = eggs, representing $2^{-\Delta\Delta Ct}$, or fold change relative to *t*' = 0. Error bars represent S.E.M. from 3–5 biological replicates, and a one-tailed *student's t*-test between agar and liquid was performed for assessing statistical differences.

## ChIP-qPCR

Nuclear fractionation, chromatin digestion, and immunoprecipitation (IP) were performed as previously described in Werner et al. 2018[40] with an additional pre-clear step prior to IP with washed, unconjugated beads for 30 min. This additional step was empirically determined to yield mildly greater enrichment *vs.* background (Input) compared to the original protocol, or an additional step of hydroxyapatite (hap) nucleosome purification. A 1M salt wash yielded greater signal *vs.* background, however it enriched for multivalent antibody binding, i.e. higher-order nucleosomes, compared to the other conditions. After pre-clear, ~10 µg of input (fully saturating beads) was used for 10-min incubations with antibody-conjugated Dynabeads (Thermo Fisher cat. #10004D) 5 µg Ab/20 µl beads, followed by five wash steps and three tube transfers. Co-precipitated DNA was purified by 0.4 mg/ml Proteinase K digestion for 2 h, 55 °C, and AMPure XP bead purification (cat. #A63881), and resuspended in 50 µl TE buffer.

Quantitative PCR (qPCR) was performed on a 1:100 dilution of co-precipitating DNA with four technical replicates, following the parameters discussed in *eud-1* qPCR (see above), and the average Ct was normalized according to the '% Input' method (Haring et al.) as follows:

$$100 * 2^{((Ct\ 10\%Input - \log2(10)) - Ct\ Ab\ sample)}$$

Statistical significance was determined by a one-way *student's t*-test (based on phenotype, WB and LC-MS/MS results).

## ChIP-Seq

ChIP-sequencing reads were aligned to the 'El Paco' genome assembly (Rödelsperger et al.) using Bowtie2 (version 2.3.4.1)(Langmead and Salzberg, 2012) with standard parameters. Mapped reads were filtered for quality (e.g. samtools view -b -q 10 'sample'.bam > 'sample'.filtered.bam), then sub-sampled to 6 million reads each per library. Metagene coverage plots were made using the bedGraph output files from MACS2 and a.bed file of the El Paco gene annotation using a custom 'awk' script to capture 5' to 3' ends of mRNA, and the Homer software package (Heinz et al.) function 'makeMetaGeneProfile.pl'. Peaks were called using Macs2 or Homer 'FindPeaks' with factor set to 'histone' on each replicate, normalized to input (both methods yielded similar peak numbers and locations). Replicate peaks within 100 bp of each other were merged with BedtoolsMerge. Tag counts at peak regions between replicates and between H4K12ac and H4K5ac were compared using 'AnnotatePeaks', and Pearson correlations were calculated in Excel. Chromatin state comparisons were done using BedTools 'IntersectBed' and previously determined genome-wide chromatin states (Werner et al. 2018[40]). A Fisher Exact test of H4K12ac peaks within regulatory regions (promoter, enhancer, and transcriptional transition) and repressed regions (repressive states 1-3) vs. expected genome-wide distributions was <0.01. Differences in H4K12ac abundance +/- TSA was performed using Macs2 bedgraph output (normalized to input) with Homer "AnnotatePeaks" with size 'given' against a peak file of gene coordinates (V3). Gene IDs with average replicate tag counts >1.5 fold-change between TSA and control were extracted. Protein sequences of these genes were obtained and homologs to *C. elegans* genes were identified with BLAST. Top hits with >25% identity, >50% positive amino acid matches and query length/sequence length >10% were input into WormBase Gene Set Enrichment Analysis[41].

## STX64 experiments

To prepare STX64 plates, we spotted 6 cm NGM-agar plates with 200 µl of OP50 + 10 µl STX64 (1mg/ml in DMSO) for a final concentration of just under 1 µg/ml. For control DMSO plates, we added 200 µl OP50 +10 µl DMSO. Wild type PS312 worms or the *eud-1* constitutive over-expression line Ex[*eud-1*](RS2561) were bleached and eggs were aliquoted to STX64 or DMSO plates. Thirty worms from STX64 plates were picked to new DMSO plates ~48 h post bleach synchronization (*n* = 3).

## Developmental delay

*P. pacificus* (PS312) and *C. elegans* (N2) were bleach-synchronized, then eggs were incubated 24 h in M9 buffer with rotation at room temperature, in order to further synchronize to the L2/J2 stage with starvation. Afterwards, animals were aliquoted into 10 ml standard S-Medium liquid cultures with 66 µM TSA (100 µl) or 100 µl DMSO, and observed at the indicated time points with a 40x oil immersion objective on an upright light microscope (Zeiss Axioscope 5). Stages were identified and scored based on vulva and mouth development, *n* = 3 for *P. pacificus* and 4 for *C. elegans*. Statistical significance was determined with a two-way ANOVA in R: aov(time ~ treatment*stage+ replicate)

*Drosophila* embryos were collected for 3h on standard grape juice agar egg-laying medium, and transferred individually to culture vials. The genotype of all flies used was *w*[1118] (Bloomington Drosophila Stock Center #3605). Flies were cultured on standard medium containing cornmeal, yeast, agar, and molasses, plus 0.1% propionic acid and 0.1% Tegosept for mold inhibition. Additionally, the medium contained either TSA dissolved in DMSO, or DMSO only as a control. These were added to the medium at a concentration of 0.34% (v/v) once it had cooled to ~50 °C. This resulted in a final concentration of 10 µM TSA in the TSA-containing medium. For each condition (10 µM TSA and DMSO-only control), four replicates were established. Each replicate consisted of a vial with 50 embryos each, for a total of 200 embryos per treatment. Beginning 118h after transfer of embryos (118h–121h post-laying), pupae and adults were counted in each vial every 24h. Flies were maintained at 25 °C and 60% humidity on a 16:8 light:dark cycle, and counting occurred 3h after lights on. Pupae were counted through Day 13 (thereafter, any new pupae could conceivably represent progeny of the original flies). Flies were sexed and counted until all replicates of both treatments stopped producing new adults, which occurred on Day 16. Each day, all new adults were frozen at −80 °C for use in Western Blots.

Statistical analysis was performed in R. Differences in time to pupation and to eclosion between TSA-treated flies and DMSO-treated controls were analyzed with two-factor ANOVA, accounting for variation between replicates (Model: time ~ replicate + treatment).

In *Drosophila* raised in media containing 10 µM TSA, the average time to pupation was 8.68 days (95% CI: 8.33–9.03) and the average time to eclosion was 12.72 days (95% CI: 12.43–13.02). In control flies treated with DMSO only, the average time to pupation was 7.35 days (95% confidence interval: 7.22–7.48) and the average time to eclosion was 11.56 days (95% CI: 11.43–11.68). Thus, flies exposed to TSA took significantly longer to develop to pupation (*p* < 0.00001, two-way ANOVA) and to eclosion (*p* < 0.00001, two-way ANOVA). The average developmental delay in TSA-treated flies was 1.33 days over the period between laying and pupation and 1.16 days over the entire life cycle from laying to eclosion, indicating that TSA delays development in the larval stages of *Drosophila* but not the pupal stage. We found that flies raised in TSA were significantly less likely to survive from laying to pupation than the DMSO-only controls (56/200 survivors in TSA vs. 142/200 in DMSO, *p* < 0.00001, two-sample chi-square test with Yates correction). However, between pupation and eclosion, the survival rate did not differ significantly between the

treatments (43/56 survivors in TSA vs. 124/142 in DMSO, $p = 0.105$, two-sample chi-square test with Yates correction), further suggesting that the main developmental effects of TSA occur prior to pupation. Consistent with previous results (Pile et al.) we found that the surviving adults in the TSA-treated group were significantly more likely to be female than DMSO-treated controls (26/43 females in TSA vs. 51/124 in DMSO, $p = 0.044$, two-sample chi-square test with Yates correction).

### Reporting summary
Further information on research design is available in the Nature Portfolio Reporting Summary linked to this article.

## Data availability
All nematode strains used in this study are available from the corresponding author Ralf J. Sommer (ralf.sommer@tuebingen.mpg.de). All other materials (oligonucleotide primers and antibodies) used in this study were purchased from vendors. The ChIP-seq datasets generated during this study are available at the National Center for Biotechnology Information Sequence Read Archive (NCBI SRA) data base under the accession number PRJNA628502. The mass spectrometry proteomics data have been deposited to the ProteomeXchange Consortium via the PRIDE partner repository with the dataset identifier PXD018940. All microscopy images are available upon request. Source data are provided with this paper.

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

## Acknowledgements

We would like to thank Silke Wahl for guidance on histone LC-MS/MS and loading digested peptides on C18 columns. We would like to acknowledge Hanh Witte and Bogdan Sieriebriennikov for creating and providing the *eud-1* CRISPR mutant. We would also like to thank Talia L. Karasov, James Lightfoot, and Tess Renahan for critical reading of our manuscript, and all members of the Sommer and Werner Laboratories. We would also like to thank WormBase. Funding was generously

provided by The Max Planck Society and the School of Biological Sciences at the University of Utah

## Author contributions

M.S.W. and R.J.S. designed experiments; M.S.W. and T.L. performed reciprocal transplant and RT-qPCR experiments. M.S.W. and T.T. performed GMM. M.S.W. extracted histones and performed nematode WBs, and prepared digested peptides for LC-MS/MS. M.F.W. performed LC-MS/MS with supervision from B.M.. T.K. performed all fly experiments, and S.R. carried out the initial STX-64 experiments. R.J.S. and M.S.W. provided resources. Writing by M.S.W. and R.J.S. with input from all authors.

## Funding

## Competing interests

The authors declare no competing interests.
