## [Peer Review File · Nature Communications]

Histone 4 lysine 5/12 acetylation enables developmental plasticity of *Pristionchus* mouth formREVIEWER COMMENTS

Reviewer #1 (Remarks to the Author):

In this manuscript, Werner et al address the mechanisms of mouth plasticity in *Pristionchus*. Building on their previous work on the role of the sulfatase gene *eud-1* in determining mouth plasticity, the authors suggest that Histone 4 lysine 5/12 in the *eud-1* locus has a permissive role in the developmental decision. The circumstantial evidence for the role of chromatin acetylation in the *eud-1* locus is strong (timing of *eud-1* expression, timing of mouth plasticity, timing of *eud-1* locus acetylation). Unless one can somehow target the acetylation of that locus specifically, the data shown is still correlational. Nevertheless, given the novelty of the role of H4K5/12ac in the phenotypic plasticity, this work merits publication.

Overall, the manuscript is well written and clear. The experiments (including the controls) are appropriate and follow the standards of the field. I would have just a few suggestions to add clarity:

- In Figure 1C, I suppose LC means Liquid Culture and AG means Agar?
- Line 43: I would add a qualifier to this statement since this was not formally shown (unless the acetylation would be solely changed in the *eud-1* gene). Maybe something like: "Entry and exit of the critical window is determined by H4K5/12ac, most likely at the switch gene *eud-1*."
- Lines 122-125: refer to the Materials and Methods for statistical tests and the number of replicates.
- Line 245: the authors cite a recent paper on the times of divergence in major Ecdysozoan groups. In that paper, there is no mention of times of divergence in nematode clades. Can the authors elaborate on where the "200 Mya" time of divergence between *C. elegans* and *P. pacificus* came from?
- Line 468: According to the Methods section, the y axis of Extended Data Fig. 1 and Fig. 5 (a-c) represents "% of Eu". Please label graphs accordingly, instead of only "%".
- Fig. 5: I find the number of replicates for nematodes (n=2) a bit low. Is there a reason for so few replicates?
- In the supplementary data for the antibodies, please mention the dilution used for each of them.
- In the supplementary data for the primers, I suppose they are oriented in the 5'→3'. If so, state it so.
- Line 656: What is γ -45? I suppose it is Ppa-Y45F10D.4 (iron-binding protein)? If so, annotate it with the full name and add the relevant reference.

Reviewer #2 (Remarks to the Author):

The manuscript by Michael S. Werner et al. try to tackle a major question in the chromatin and developmental biology fields: how the environment impacts chromatin and ultimately cell fate. The authors used a well-established nematode model system, *Pristionchus pacificus*, that shows mouth dimorphism. By employing this very well tractable system, the authors showed that the histone modification H4K12ac responds to environmental clues allowing a permissive chromatin environment at specific switch genes. Moreover, the authors showed that H4K12ac also carries epigenetic memories of environmental exposure and that the memory can be affected by using TSA mediated HDAC inhibition.

The work is very sound and would definitely deserve publication in Nature Communications. I nevertheless suggest here some comments to the authors with the hope that a few further experiments would improve and solidify the major claims of the study.

After 36h, the authors found that the levels of *eud-1* start to decrease. Moreover, constitutive overexpression of *eud-1* gives 100% Eu animal.

I wonder whether overexpression of *eud-1* from 36h onwards and/or providing the steroid sulfatase in the media would yield Eu animals.

The authors suggest that in physiological conditions H4K12ac is removed from the eud-1 gene by the action of HDACs resulting in more compact chromatin.

It would be interesting to confirm this by, for example, chipping H3K27me3 and see whether now there is a competitive enrichment for this repressive histone mark on the eud-1 gene. Another option to study the interplay with H3K27me3 could be addressed by using mutants of mes-2, although I am not sure if this mutant is available in *Pristionchus pacificus*. On the market, there is also a potent inhibitor of EED that is the homologue of mes-6 and that could be an option to prevent H3K27me3 accumulation.

Minor:

Line 248: "in 2001" not in 2003

The authors suggested that elongation is important and observed changes in eud-1 transcripts level from bulk total RNA. It would be nice to check for primary transcripts instead of mRNA. For example, the authors could design oligos on introns and confirm that the eud-1 increased levels are indeed the result of de-novo transcription and not mRNA increased stability.

Reviewer #3 (Remarks to the Author):

NCOMM Sommer

This is a creative and interesting paper that needs some adjustments. The authors explore the effects of the environment on the teeth of *Pristionchus pacificus*. Their major findings are convincingly shown, as they list out:

1. Reciprocal transplant experiments reveal a critical time window of mouth-form plasticity.
2. Entry and exit of the critical window is determined by H4K5/12ac at the switch gene eud-1
3. H4K12ac maintains transcriptional competence by supporting elongation.
4. Inhibition of deacetylation freezes an initial developmental trajectory, resulting in long-term epigenetic memory.
5. H4K5/12 acetylation control of plasticity was co-opted from an ancestral role in controlling developmental speed

The authors see a convincing effect on developmental choice by histone acetylation drugs. Production of Eu mouths by TSA, which blocks HDACs and leads to higher histone acetylation, particularly H4Ac. Treatment alters the environmental response to liquid culture and prolongs the temperature sensitive period past 36 hours. The implication is that deacetylation helps terminate the plasticity window. The effect seems fairly specific for H4K12 and/or H4K5, and not other acetylation. TSA induced eud-1 expression, the master regulator of the eu phenotype. Environmental responses are an exciting but ill-understood field. This paper has the possibility of showing how modified histone contribute to one environmental response.

1. TSA slows development, which could account for the effects they see on mouth development. The authors should perform more controls to separate developmental choice from developmental delay. First, if they mutate candidate acetylases or deacetylases for K12Ac, do they recapitulate the effects on mouth choice? On development?. Separating mouth choice from a general developmental delay is critical. And perhaps, add the TSA after the general developmental delay but right before the mouth extension choice (36 hours or later) to see if this separates the effects.
2. Is eud-1 required for the HSA or acetylase effect? The authors show a small effect on acetylation of

eud-1, so it's important to know if it matters by making the double combination. The authors should include more about the genome-wide chip seq (figure 3) and what other genes or families of genes are altered. What are the top hits.

3. Code: the authors state in their abstract and text that there is a histone code for plasticity. This is over-interpretation of the data, and should be cut. There is no evidence for a code.

Figure 1: The teeth are impossible to see. A cartoon near the images could help non experts see the relevant structures.

Figures: the yellow barely shows on the white page. Could the authors make it a different color.

Response to reviewers:

Reviewer #1:

*“In this manuscript, Werner et al address the mechanisms of mouth plasticity in *Pristionchus*. Building on their previous work on the role of the sulfatase gene *eud-1* in determining mouth plasticity, the authors suggest that Histone 4 lysine 5/12 in the *eud-1* locus has a permissive role in the developmental decision. The circumstantial evidence for the role of chromatin acetylation in the *eud-1* locus is strong (timing of *eud-1* expression, timing of mouth plasticity, timing of *eud-1* locus acetylation). Unless one can somehow target the acetylation of that locus specifically, the data shown is still correlational. Nevertheless, given the novelty of the role of H4K5/12ac in the phenotypic plasticity, this works merits publication. Overall, the manuscript is well written and clear. The experiments (including the controls) are appropriate and follow the standards of the field.”*

- We thank the reviewer for the positive evaluation of our manuscript.

“In Figure 1C, I suppose LC means Liquid Culture and AG means Agar?”

- Indeed. We have changed the legend to reflect the full name and thereby prevent any confusion.

*“Line 43: I would add a qualifier to this statement since this was not formally shown (unless the acetylation would be solely changed in the *eud-1* gene). Maybe something like: “Entry and exit of the critical window is determined by H4K5/12ac, most likely at the switch gene *eud-1*.”*

- We agree and have made the appropriate change.

“Lines 122-125: refer to the Materials and Methods for statistical tests and the number of replicates.”

- We agree and have made the appropriate change.

*“Line 245: the authors cite a recent paper on the times of divergence in major Ecdysozoan groups. In that paper, there is no mention of times of divergence in nematode clades. Can the authors elaborate on where the “200 Mya” time of divergence between *C. elegans* and *P. pacificus* came from?”*

- In the reference Howard et al., 2022, figure 4 shows a phylogeny of Ecdysozoan lineages with posterior age estimates. Some of these estimates are even rooted by fossil evidence. *Pristionchus* and *Caenorhabditis* exhibit a 95% higher posterior density range centered around 200 million years ago. We believe this paper is the most substantial investigation of Ecdysozoan evolution, and therefore thought it is appropriate of being cited. Nevertheless, to be conservative, we have modified the text to include a range of time, beginning from the more commonly accepted ~100 million years.

“Line 468: According to the Methods section, the y axis of Extended Data Fig. 1 and Fig. 5 (a-c) represents “% of Eu”. Please label graphs accordingly, instead of only “%.”

- These graphs represent % of total developmental stage, and have been updated to reflect that.

“Fig. 5: I find the number of replicates for nematodes (n=2) a bit low. Is there a reason for so few replicates?”

- Indeed - the ‘n=2’ was from an earlier version of the manuscript before we added another replicate for *Pristionchus* and two more for *C. elegans*. The data that was presented was the updated version, but regrettably we forgot to change the ‘n’ in the methods section. We thank the reviewer for pointing this out. We also added a significance test (two-way ANOVA) in both nematode species to further demonstrate the developmental delay caused by TSA.

- In the supplementary data for the antibodies, please mention the dilution used for each of them.

- We agree and have made the appropriate change in an updated version of Extended Data Table 1.

“In the supplementary data for the primers, I suppose they are oriented in the 5’->3’. If so, state it so.”

- We agree and have made the appropriate change in an updated version of Extended Data Table 1.

“Line 656: What is y- 45? I suppose it is Ppa-Y45F10D.4 (iron-binding protein)? If so, annotate it with the full name and add the relevant reference.”

- We agree and have made the appropriate change in the methods section.

Reviewer #2:

“The work is very sound and would definitively deserve publication in Nature Communications. I nevertheless suggest here some comments to the authors with the hope that a few further experiments would improve and solidify the major claims of the study.”

- We thank the reviewer for their positive comments and appreciate their suggestions to improve our manuscript.

“After 36h, the authors found that the levels of eud-1 start to decrease. Moreover, constitutive overexpression of eud-1 gives 100% Eu animal. I wonder whether overexpression of eud-1 from 36h onwards and/or providing the steroid sulfatase in the media would yield Eu animals.”

- This is a great idea as it gets to the heart of the mechanism of TSA’s effect on mouth form. Unfortunately, in its exact form it is not technically possible as we do not have inducible expression systems in *P. pacificus*, largely due to the difficulty of inserting large fragments (>100 bp) and the fact that extrachromosomal arrays are almost

always ephemeral in *P. pacificus*. *However*, we thought of another experiment that we believe is in the same spirit as the reviewer's suggestion. By culturing a *eud-1* overexpression line in the presence of the sulfatase inhibitor STX-64, and then switching during the critical window to media without the inhibitor, we should effectively test the same point: does the presence of EUD-1 *after* the critical window induce the Eu morph, similar to the effect seen by TSA? We have conducted this experiment and are pleased to report that EUD-1 activity after 48 hours does indeed induce the Eu morph. These results, which are also the first use of STX-64 in *P. pacificus*, are now presented in **Extended data Figure 7b-c**, and described in the text at the end of the "An H4 acetylation/deacetylation timer determines the critical window" results section. We feel that this experiment strongly supports our interpretation of the TSA data, and thank the reviewer for inspiring us to add it.

"The authors suggest that in physiological conditions H4K12ac is removed from the eud-1 gene by the action of HDACs resulting in more compact chromatin.

It would be interesting to confirm this by, for example, chipping H3K27me3 and see whether now there is a competitive enrichment for this repressive histone mark on the eud-1 gene. Another option to study the interplay with H3K27me3 could be addressed by using mutants of mes-2, although I am not sure if this mutant is available in Pristionchus pacificus. On the market, there is also a potent inhibitor of EED that is the homologue of mes-6 and that could be an option to prevent H3K27me3 accumulation."

- We agreed that this model was worth investigating. To this end, we conducted two experiments. First, we performed ChIP-seq for H3K27me3 during the critical window (48 hours; the same time point and culture condition as the histone acetylation data). Second, we performed quantitative ChIP-qPCR for H3K27me3 before (24 hours) and after (72 hours) the critical window, which we assumed would provide the greatest difference in heterochromatin abundance.

In the ChIP-seq data, we observed a broad distribution of H3K27me3 across the super-gene locus that bordered - and appeared to be insulated by - the peaks of histone acetylation. However, in contrast to H4K12ac, we did not observe a significant difference in H3K27me3 abundance at different developmental time points by ChIP-qPCR. Together, these data indicate that the supergene locus is in a *facultative* heterochromatin environment throughout development. More quantitative methods such as ICE-ChIP (Grzybowski et al., 2015), or single-cell approaches, may reveal more subtle increases in H3K27me3 over time. Alternatively, probing different heterochromatin marks, such as H3K9me3, may reveal a transition from facultative to *constitutive* heterochromatin before and after the critical window.

Nevertheless, these data provide three new conclusions: First, facultative heterochromatin is present over the super-gene locus. Second, a test of the hypothesis that histone acetylation and H3K27me3 are competition with each other (which does not appear to be the case). Third, the generation of a new hypotheses: a potential role of histone acetylation in insulating heterochromatin spread – an intriguing possibility given that nematodes lack the canonical insulator protein CTCF. It will also spur the investigation of additional heterochromatin marks such as H3K9me3. Thus, we thank the reviewer for motivating us to do these experiments, which we feel substantially

strengthens our manuscript. These new data are presented in new Extended Data Fig. 6.

Regarding inhibitors of the PRC2 complex, we note that the EZH2 inhibitor (GSK343) had not exhibited a discernable effect on mouth form ratios in our previous assay (Fig. 2). Additionally, we have found, to our surprise, that several components of the PRC2 complex, as well as some other histone-modifying enzymes, have undergone rapid gain/loss in *P. pacificus*. This is the subject of another project in the first author's new lab, but likely precludes the usefulness of the suggested experiment for this manuscript.

“Line 248: “in 2001” not in 2003”

- We have made the appropriate correction and thank the reviewer for pointing it out.

“The authors suggested that elongation is important and observed changes in eud-1 transcripts level from bulk total RNA. It would be nice to check for primary transcripts instead of mRNA. For example, the authors could design oligos on introns and confirm that the eud-1 increased levels are indeed the result of de-novo transcription and not mRNA increased stability.”

- We felt that this was an excellent suggestion, and conducted a new experiment to address this point. Reverse transcription was performed with both oligo dT to capture mRNA, and a gene-specific primer that anneals to the third intron to capture primary transcripts (from a 48 hour time point +/- TSA). qPCR was then performed as previously (n=3). As expected, the increase in *eud-1* mRNA corresponds with an increase in primary *eud-1* transcription. These data are now presented as a new supplementary figure (Extended Data Fig. 3d).

Reviewer #3:

*“This is a creative and interesting paper that needs some adjustments. The authors explore the effects of the environment on the teeth of *Pristionchus pacificus*...Environmental responses are an exciting but ill-understood field. This paper has the possibility of showing how modified histone contribute to one environmental response.”*

- We thank the reviewer for appreciating the impact of our work to environmental influence and their thoughtful suggestions to improve it.

“1. TSA slows development, which could account for the effects they see on mouth development. The authors should perform more controls to separate developmental choice from developmental delay. First, if they mutate candidate acetylases or deacetylases for K12Ac, do they recapitulate the effects on mouth choice? On development?. Separating mouth choice from a general developmental delay is critical. And perhaps, add the TSA after the general developmental delay but right before the mouth extension choice (36 hours or later) to see if this separates the effects.”

- This is an important point, and we have addressed the reviewer's comment in five ways:

(1) First, we generated five mutant HDAC's (out of 15 total) using CRISPR-Cas9 (Revision Figure 1). Two appear to be homozygous lethal, as heterozygote mothers produced 0/16 homozygous mutant progeny. We were able to obtain three homozygous HDAC mutants with frameshift mutations, however none yielded an observable effect on mouth form (from three or more different mutant alleles). There are 10 HDACs remaining, however we believe a full knockout panel of all *P. pacificus* HDACs, and functional characterization, is meritorious for a subsequent manuscript and thus beyond the scope of this manuscript. HDACs are also known to exhibit substrate redundancy (Seto and Yoshida, 2014), which may make it challenging to find a 1:1 phenotype attributable to TSA.

(2) We noted that animals treated with the histone acetyltransferase drug NU9056 (Figure 2a) exhibit delayed development. Interestingly, we have also anecdotally observed that mutant strains of the acetyltransferase *Ppa-lsy-12* (Serobyán et al., 2016) exhibit delayed development. To quantify this difference, we measured developmental stages every 12 hours after bleach-synchronization (Revision Figure 2). Indeed, *Ppa-lsy-12* mutants are developmentally delayed compared to wild type worms. Importantly, both *Ppa-lsy-12* mutants and NU9056 increase the percentage of St animals. These data support a role for acetylation in morph choice, but also separate the direction of morph choice from developmental speed. It's also worth mentioning that although wild-type St worms on standard conditions develop ~6 hours faster than Eu animals, most treatments that increase St also slow development: cold temperature (Lenuzzi et al., 2022), excess ions (Ragsdale et al., 2013), DMSO (Werner et al., 2017), and ethanol (unpublished). Thus, we believe that delayed development and Eu induction observed in +TSA conditions are a specific, regulatory effect, and not caused by a general developmental delay.

(3) We have included an additional sentence in the Results section describing our perspective, which is that developmental delay and induction of the Eu morph are mechanistically linked – and are thus not mutually exclusive.

(4) The candidate H4K12-acetyltransferase in *Drosophila (Chamaeu)* does not have an obvious homolog in *P. pacificus* (see section above from Reviewer #2 on PRC2), but identifying which acetyltransferase is writing that mark in *Pristionchus* is an on-going research project in the first author's new laboratory. This requires systematic analysis and will involve the inactivation and characterization of more than a dozen additional genes.

(5) Finally, regarding the suggestion to add TSA after developmental delay: This experiment would be difficult to interpret for the purpose of separating delay from morph choice. From our staging experiments, we see that at 44 hours, +/- TSA conditions are developing at the same speed (Figure 5a), and only begin to separate at 48 hours. This time period coincides with the end of the critical window, so any addition of TSA will have diminishing effects on morph choice. Indeed, in Figure 4 we have a +TSA-at-48 hours condition that was used as a control. As expected, this experiment did yield an increase in %Eu compared to DMSO control, but with a smaller effect size than adding TSA at t'0 – consistent with closing the critical window.

b

P. pacificus	H. sapiens homolog	C. elegans homolog	injected P0s	RFP+ (co-injection) F1 plates	heterozygote F2s isolated	homozygous mutant F3s	frame shift?	mutant allele	Strain ID
isopolya.174.2.2	HDAC-1	hda-1	30	2	2	0			
PPA17629	HDAC-1	hda-1	30	2	3	1. 16/14-12: 14 bp deletion 2. 16/6-5: 6 bp 3. 19/30-3: 7 bp 4. 16/19-7: 13 bp insertion, 3 bp	[x] [x] [x] [x]	tu1586 tu1587 tu1585 tu1578	RS3813 RS3814 RS3812 RS3808
DN28546_c0_g1_i2	HDAC-1	hda-3	30	2	6	1. 1/13-6: 8 bp deletion 2. 1/3-8: 37 bp insertion, 2 bp 3. 2/4-3: 8 bp	[x] [x] [x]	tu1579 tu1581 tu1580	RS3809 RS3811 RS3810
DN28176_c1_g1_i1	HDAC-3	hda-2	30	5	10	0			
PPA00049	HDAC4,5,7,9	hda-4	30	1	7	19-3: 11 bp deletion 27-1: 5 bp deletion 10-4: 25 bp insertion + 3 bp	[x] [x] [x]	tu1583 tu1582 tu1584	RS3830 RS3829 RS3831

Revision Figure 1: a) Phylogenetic tree of *P. pacificus* HDACS. Corresponding homologous HDAC families are indicated by branch color, and CRISPR mutants are indicated with an 'x' **b)** Table of CRISPR mutants, homologs in humans and *C. elegans*, and mutant allele and strain ID numbers.

Revision Figure 2: Stage- screen of Ppa-lsy-12 and wild type (PS312) worms after hypochlorite synchronization (n=3). Earlier stages did not show noticeable differences in development.

“2. Is *eud-1* required for the HSA or acetylase effect? The authors show a small effect on acetylation of *eud-1*, so it’s important to know if it matters by making the double combination.”

- We thought this was an excellent suggestion and conducted an experiment with TSA in a *eud-1* mutant background. The result was all St (n=3), demonstrating that *eud-1* is required for TSA’s effect. These data are now featured in the Results section text (pg. 13).

“The authors should include more about the genome-wide chip seq (figure 3) and what other genes or families of genes are altered. What are the top hits.”

- We have included a more thorough analyses of histone acetylation ChIP data in a new supplemental figure (Extended Data Fig. 5). Specifically, we assessed the correspondence between H4K5ac and H4K12ac, genome-wide-H4K12ac peak location, and Gene Set Enrichment Analysis of genes hyperacetylated for H4K12ac in the presence of TSA (>1.5 fold change).
 - The results are as follows: H4K5ac and H4K12ac show a strong correlation (Pearson corr. = 0.82, 0.84 for replicates 1 and 2, respectively).
 - H4K12ac peaks were significantly enriched in regulatory regions when compared with a previously annotated chromatin state map (p<0.01, Fisher’s Exact Test)(Werner et al., Genome Res., 2018).
 - Finally, genes hyperacetylated by TSA have both Gene Ontology (GO) and RNAi phenotype-enrichment that corresponds with *cell proliferation*. This is an exciting result to us because one of the hypotheses for why TSA has chemotherapeutic effects is by inhibiting cell proliferation. Our data support this hypothesis and suggest a deeply conserved role for acetylation in controlling cell proliferation/apoptosis. We have included a description of these analyses in a new Results section: ‘*Genome-wide patterns of H4 acetylation and a role in switch-gene expression*’. Future investigations in the first author’s lab will explore this link, which based on our biochemical data, we hypothesize is mediated by H4K5/12ac. We thank the reviewer for motivating us to do these analyses which provide more depth to the ChIP-Seq data and a new angle to our manuscript.

“3. Code: the authors state in their abstract and text that there is a histone code for plasticity. This is over-interpretation of the data, and should be cut. There is no evidence for a code.”

- We concede that the ‘code’ was speculative and have followed the advice to remove it.

Figure 1: The teeth are impossible to see. A cartoon near the images could help non experts see the relevant structures.

- We thank the reviewer for pointing this out and have included a cartoon image below the microscopy images.

“Figures: the yellow barely shows on the white page. Could the authors make it a different color.”

- We suspect that the yellow the reviewer is referring to is in the agar and liquid culture cartoons (?). If so, we suspect that the low-resolution images embedded within the manuscript are the cause of the problem, which should be solved by high resolution images. Thus, we prefer to leave the tint for now, but would be more than happy to change it, if it continues to be an issue.

References:

Lenuzzi, M., Witte, H., Riebesell, M., Rödelsperger, C., Hong, R.L., and Sommer, R.J. (2022). Influence of environmental temperature on mouth-form plasticity in *Pristionchus pacificus* acts through daf-11-dependent cGMP signaling. *J Exp Zoology Part B Mol Dev Evol* <https://doi.org/10.1002/jez.b.23094>.

Ragsdale, E.J., Müller, M.R., Rödelsperger, C., and Sommer, R.J. (2013). A Developmental Switch Coupled to the Evolution of Plasticity Acts through a Sulfatase. *Cell* *155*, 922–933. <https://doi.org/10.1016/j.cell.2013.09.054>.

Werner, M.S., Sieriebriennikov, B., Loschko, T., Namdeo, S., Lenuzzi, M., Dardiry, M., Renahan, T., Sharma, D.R., and Sommer, R.J. (2017). Environmental influence on *Pristionchus pacificus* mouth form through different culture methods. *Sci Rep-Uk* *7*, 7207. <https://doi.org/10.1038/s41598-017-07455-7>.

REVIEWERS' COMMENTS

Reviewer #2 (Remarks to the Author):

The revised manuscript by Michael S. Werner et al. is now extraordinarily compelling and deserves a speedy publication. The authors addressed all the points I raised painstakingly.

Reviewer #3 (Remarks to the Author):

Werner et al. resubmission

The authors have responded to our comments and the paper is improved and almost ready to go. Two small adjustments:

The ChIP differences are fairly subtle in Figure 3, could the authors provide in the figure or supplement the z scores as a y axis rather than % input.

In figure 2i could the authors include the fold increase for H4K16Ac, since that clearly rises also.

Point-by-point response:

REVIEWERS' COMMENTS

Reviewer #2 (Remarks to the Author):

The revised manuscript by Michael S. Werner et al. is now extraordinarily compelling and deserves a speedy publication. The authors addressed all the points I raised painstakingly.

We thank the reviewer for their positive evaluation of our revised manuscript.

Reviewer #3 (Remarks to the Author):

Werner et al. resubmission

The authors have responded to our comments and the paper is improved and almost ready to go. Two small adjustments:

The ChIP differences are fairly subtle in Figure 3, could the authors provide in the figure or supplement the z scores as a y axis rather than % input. In figure 2i could the authors include the fold increase for H4K16Ac, since that clearly rises also.

We thank the reviewer for their positive evaluation of our revised manuscript.

- We have included the fold-increase for H4K16ac (LC-MS/MS data) in Fig. 2i
- We have added a new figure panel to Extended Data Fig. 6 (c) that presents the z-scores of the ChIP-qPCR data presented in Fig. 3.